# Less Could Be Better: Parameter-efficient Fine-tuning Advances Medical Vision Foundation Models

**Chenyu Lian**[1]                                                     CYLIAN@STU.XMU.EDU.CN
[1] *School of Informatics, Xiamen University*

**Hong-Yu Zhou**[2]                                          WHUZHOUHONGYU@GMAIL.COM
[2] *Department of Biomedical Informatics, Harvard University*

**Yizhou Yu**[3]                                                          YIZHOUY@ACM.ORG
[3] *Department of Computer Science, The University of Hong Kong*

**Liansheng Wang**[*1]                                               LSWANG@XMU.EDU.CN

**Editors:** Under Review for MIDL 2024

## Abstract

Parameter-efficient fine-tuning (PEFT) that was initially developed for exploiting pre-trained large language models has recently emerged as an effective approach to perform transfer learning on computer vision tasks. However, the effectiveness of PEFT on medical vision foundation models is still unclear and remains to be explored. As a proof of concept, we conducted a detailed empirical study on applying PEFT to chest radiography foundation models. Specifically, we delved into LoRA, a representative PEFT method, and compared it against full-parameter fine-tuning (FFT) on two self-supervised radiography foundation models across three well-established chest radiograph datasets. Our results showed that LoRA outperformed FFT in 13 out of 18 transfer learning tasks by at most 2.9% using fewer than 1% tunable parameters. Combining LoRA with foundation models, we set up new state-of-the-art on a range of data-efficient learning tasks, such as an AUROC score of 80.6% using 1% labeled data on NIH ChestX-ray14. We hope this study can evoke more attention from the community in the use of PEFT for transfer learning on medical imaging tasks. Code and models are available at `https://github.com/RL4M/MED-PEFT`.

**Keywords:** Transfer learning, Medical vision foundation models, Chest X-ray.

## 1. Introduction

Full-parameter fine-tuning (FFT) has long been recognized and adopted as a superior technique to do transfer learning (He et al., 2022; Wang et al., 2023; Zhou et al., 2023a,c; Yu et al., 2020). However, foundation models usually have a large number of parameters, and fine-tuning the full model weights can be a sub-optimal choice when the downstream task only has limited annotations. This contrast deserves more attention in medical imaging tasks where annotation is often hard to access due to issues like privacy and safety and also the rare nature of certain diseases. On the other hand, parameter-efficient fine-tuning (PEFT) (Houlsby et al., 2019; Hu et al., 2021; Liu et al., 2022) was proposed to largely reduce the number of model parameters to be tuned and has been widely used in both language (He et al., 2021; Zhang et al., 2023; Ponti et al., 2023) and vision tasks (Jia et al., 2022; Sung et al., 2022; Yang et al., 2023).

---

[*] Corresponding author

Table 1: Comparison of the classification results of FFT and LoRA on MAE and MRM, while 1%, 10%, and 100% denote the ratios of labeled data used for fine-tuning.

| Pre-trained Models | Transfer Methods | NIH | | | CheXpert | | | RSNA | | |
|---|---|---|---|---|---|---|---|---|---|---|
| | | 1% | 10% | 100% | 1% | 10% | 100% | 1% | 10% | 100% |
| MAE | FFT | 74.2 | 82.2 | 85.6 | 87.3 | 90.3 | 91.8 | 89.6 | 90.5 | 93.1 |
| | LoRA | 77.1 | 82.9 | 85.7 | 88.4 | 91.1 | 91.1 | 89.9 | 91.9 | 93.3 |
| | | (+2.9) | (+0.7) | (+0.1) | (+1.1) | (+0.8) | (-0.7) | (+0.3) | (+1.4) | (+0.2) |
| MRM | FFT | 80.1 | 84.1 | 85.9 | 90.5 | 91.5 | 91.6 | 91.3 | 92.8 | 93.3 |
| | LoRA | 80.6 | 84.0 | 85.8 | 90.7 | 92.0 | 91.5 | 91.2 | 93.1 | 93.5 |
| | | (+0.5) | (-0.1) | (-0.1) | (+0.2) | (+0.5) | (-0.1) | (-0.1) | (+0.3) | (+0.2) |

More recently, some studies tried applying PEFT for medical image analysis (Dutt et al., 2023; Zhu et al., 2023). However, one limitation of these work is that they only investigated ImageNet (Deng et al., 2009) pre-trained models and ignored the more generalizable vision foundation models that were trained on large-scale medical data with self-supervised learning (Zhou et al., 2023b; Jiang et al., 2023). In this paper, we focus on LoRA (Hu et al., 2021), a representative PEFT method, comparing it to FFT on two self-supervised radiography foundation models across three well-established chest radiograph datasets. Experimental results indicate that in 13 out of 18 transfer learning tasks, LoRA exhibits superior performance over FFT, sometimes by notable margins. For instance, on the NIH ChestX-ray dataset with merely 1% labeled data, LoRA outperforms FFT by 2.9% with only 0.3% tunable parameters.

## 2. Experiments and Analyses

### 2.1. Settings

**Datasets.** Three chest radiograph datasets were adopted to evaluate the performance of transfer learning, including NIH ChestX-ray (NIH) (Wang et al., 2017), CheXpert(Irvin et al., 2019), and RSNA pneumonia (RSNA) (Shih et al., 2019). To analyze the data efficiency of different fine-tuning methods, we also presented results with different labeling ratios. We employed the same data splits and evaluation metrics as of (Zhou et al., 2023a) except that we used the official test set instead of the validation set of CheXpert.
**Chest Radiography Foundation Models.** We adopted two self-supervised foundation models, MRM (Zhou et al., 2023a) and MAE (He et al., 2022). Both of them were pre-trained on the MIMIC-CXR (Johnson et al., 2019) dataset, based on which LoRA and FFT were applied and compared.

### 2.2. Effectiveness of LoRA

Table 1 compares the classification results of FFT and LoRA based on MAE and MRM, measured by AUROC (%). Improvements can be observed in 13 out of 18 tasks, manifesting the universality of LoRA on different radiography foundation models and datasets. Moreover, the outstanding performance of LoRA on 1% and 10% labeled data indicates its high data efficiency, which is particularly meaningful for medical imaging limited by the scarcity of data. On 100% labeled data, LoRA performs competitively with FFT but by tuning only 1.5% parameters, showing the efficiency in computation and storage.

Table 2: LoRA ranks analysis.

| LoRA Rank | 2 | 4 | 8 |
|---|---|---|---|
| 1% | 80.4 | **80.6** | 80.4 |
| LoRA Rank | 8 | 16 | 32 |
| 10% | 84.0 | **84.1** | 84.0 |
| LoRA Rank | 16 | 32 | 64 |
| 100% | 85.7 | **85.8** | 85.8 |

Table 3: Comparison of pre-training epochs.

| Methods | Epochs of Pretraining | AUROC (%) |
|---|---|---|
| FFT | 100 | 74.4 |
|  | 200 | 74.2 (-0.2) |
| LoRA | 100 | 75.9 |
|  | 200 | 77.1 (+1.2) |

## 2.3. Ablation Analyses

**LoRA Rank Analysis.** We compare the performances of different ranks of LoRA on 1%, 10%, and 100% labeled data of NIH based on MRM, showing that the ranks of LoRA should be increased accordingly as the data scale. AUROC (%) scores are reported in Table 2.

**Pre-training Epochs Analysis.** Pre-training was conducted on the MIMIC-CXR dataset using MAE (He et al., 2022) for 100 and 200 epochs. As shown in Table 3, 1.2% improvement on 1% labeled data of NIH is observed when the pre-training epochs are extended from 100 to 200, while no improvement is witnessed for FFT. We hypothesize that LoRA benefits from the small number of tuned parameters (0.3%), mitigating the catastrophic forgetting.

## 2.4. More Analyses on Other Vision Foundation Models

**Scaling up the Foundation Models.** We conducted MAE pre-training using MIMIC-CXR images on ViT-Large (Dosovitskiy et al., 2020) for 200 epochs. Table 4 shows the further improvements when scaling up the transformer network. It is noteworthy that the result of LoRA based on ViT-Base is even 0.9% higher than the one of full-parameter fine-tuning on ViT-Large, and when adopting LoRA on ViT-Large, the AUROC of NIH 1% can be further promoted to 77.7%.

**Fine-tuned on Natural Images Pre-trained Foundation Models.** The results on Table 5 show that when adopting the natural images pre-trained models Dinov2 (Oquab et al., 2023) by FFT, the performance is substantially below the baseline. While showing that ChestX-ray pre-training is still necessary to ensure downstream performance, the introduction of LoRA significantly mitigates the performance gap caused by different modalities.

Table 4: Comparison of model scales.

| Method | ViT Scale | AUROC (%) |
|---|---|---|
| FFT | Base | 74.2 |
|  | Large | 76.2 (+2.0) |
| LoRA | Base | 77.1 (+2.9) |
|  | Large | 77.7 (+3.5) |

Table 5: On natural foundation models.

| Method | Model | AUROC (%) |
|---|---|---|
| FFT | MAE ViT-B16 | 74.2 |
| FFT | Dinov2 ViT-B14 | 66.6 (-7.6) |
|  | Dinov2 ViT-L14 | 70.9 (-3.3) |
| LoRA | Dinov2 ViT-B14 | 70.3 (-3.9) |
|  | Dinov2 ViT-L14 | 72.5 (-1.7) |

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
