# OpenReview forum: "Less Could Be Better: Parameter-efficient Fine-tuning Advances Medical Vision Foundation Models"
_MIDL.io/2024/Short_Papers — MIDL 2024 Short Papers_

### Official Review · Reviewer_pow5 · 2024-04-17

**Confidence:** 5
**Final Rating:** 3.5

**Review:**

This work explores using LoRA for Parameter-efficient fine-tuning (PEFT) of MAE and MRM networks. The method is evaluated on three large-scale chest X-ray images. Although the concept of LoRA is not new its investigation for MAE fine-tuning is interesting. Would be interesting to see if the idea holds for other medical imaging datasets. Should be accepted for presentation at the MIDL conference.

---

### Decision · Program_Chairs · 2024-04-26

Accept